# Sexual Violence against Adolescents in the State of Espírito Santo, Brazil: An Analysis of Reported Cases

**DOI:** 10.3390/ijerph192114481

**Published:** 2022-11-04

**Authors:** Mayara Alves Luis, Franciéle Marabotti Costa Leite, Nicole Letourneau, Nátaly Adriana Jiménez Monroy, Luciana Graziela de Godoi, Luís Carlos Lopes-Júnior

**Affiliations:** 1Graduate Program in Public Health, Health Sciences Center, Federal University of Espírito Santo (UFES), Vitoria 29047-105, Brazil; 2Faculty of Nursing, University of Calgary, Calgary, AB T2N 4V8, Canada; 3Nursing Department, Federal University of Espírito Santo (UFES), Vitoria 29075-910, Brazil

**Keywords:** domestic violence, adolescents, sexual violence, health information systems, epidemiology

## Abstract

Objective: We describe the prevalence of the reported cases of sexual violence against adolescents and analyze their associated factors. Methods: A cross-sectional analytical study (*n* = 561) was conducted with reported data on sexual violence against adolescents in the state of Espírito Santo registered in SINAN between 2011 and 2018 to understand the prevalence and predictors of sexual violence against adolescent victims, as well as to describe the perpetrators and the nature of the aggression. Variables to characterize the victim, aggression, and perpetrator were used. Bivariate analyses were performed using chi-square (χ^2^) and Fisher’s exact tests, and multivariate analyses were conducted using log-binomial models; the results were presented with prevalence ratios. All analyses were stratified by sex. Results: The prevalence of sexual violence was 32.6%, and 93% of the victims were female. In both males and females, the reported sexual violence was associated with a younger age (10–12 years old), living at home, being related to the perpetrator, and a history of sexual violence. In females, the reported sexual violence was also associated with the number of perpetrators, and in males, with the perpetrator’s age. Conclusions: Our findings show the high frequency of reporting of sexual violence and the characteristics of the victim, the aggression, and the aggressor as factors associated with its occurrence in both sexes. The importance of health information systems for disseminating data and the need for measures to prevent and treat the violence among adolescents is urgent.

## 1. Introduction

Adolescence is a stage of vulnerability to sexual violence, whether in the domestic or community context, especially in the case of girls, affecting youth of all levels of income, development, ages, and countries [1]. Within the home, most cases are perpetrated by relatives or acquaintances of the victims [1,2]. In the community context, the risks facing the occurrence of sexual violence increase as adolescents begin to attend new settings, where there is greater contact with groups of peers and involvement in affective relationships [1,3].

According to data from the United Nations Children’s Fund (UNICEF) [1], around 15 million adolescent girls between 15 and 19 years old have experienced forced sex during their lifetime. In a survey conducted in 20 countries, approximately 9 out of 10 victims of sexual violence reported that the act occurred for the first time during adolescence. A systematic review and meta-analysis study suggested that the prevalence of child sexual violence worldwide is approximately 20% among girls and 8% among boys [4].

Data on sexual violence in Brazil against children and adolescents are alarming. According to the Brazilian National Human Rights Ombudsman, 17,073 reports of sexual violence were made through Dial 100, a telephone channel for anonymous reporting of acts that violate human rights in Brazil [5]. Between 2011 and 2017, 184,524 suspected or confirmed cases of sexual violence were registered in the Information System for Reporting Diseases (SINAN—Sistema de Informação de Agravos de Notificação), with 40.5% of these cases occurring against adolescents [6]. A study conducted in the state of Santa Catarina demonstrates that the prevalence of reporting of sexual violence is higher in adolescents when compared to occurrence in women over 20 years of age [2]. Still, the Brazilian National Survey of School Health (PeNSE—Pesquisa Nacional de Saúde do Escolar) showed that the prevalence of sexual violence among students enrolled in elementary school in Brazil, in 2015, was 4.0% [7]. Regardless of the type, every act of sexual violence is both intrusive and traumatic, ranging from direct physical contact to the use of force, or occurring through less direct forms, such as unwanted exposure to sexual language and images [8].

This phenomenon has short- to long-term impacts on adolescents’ physical and mental health, such as sexually transmitted infections, early pregnancy, gastrointestinal disorders [9,10,11,12], post-traumatic stress disorder, anxiety, depression, eating and sleep disorders, relationship problems, and suicidal behaviors [11,13,14,15,16]. In addition to these consequences, studies also show engagement in risky sexual behavior [17,18] and substance abuse [19,20] as a result of sexual violence. In this context, the protection of youth against all forms of violence is a fundamental right guaranteed by international treaties and human rights standards [21]. The Convention on the Rights of the Child, held in 1989, had an impact on national and international legislation, programs, and policies in favor of the promotion and protection of children’s rights.

In Brazil, until the advent of the Federal Constitution of 1988, children and adolescents were not considered subjects of rights. Since then, modern legislation has been adopted in accordance with the United Nations Convention on the Rights of the Child, in order to ensure the rights of those under eighteen years of age [22]. As the main normative instrument in Brazil with regard to this population, in 1990 the Children and Adolescents Statue was sanctioned to ensure full protection for these individuals [23].

In the field of health, compulsory reporting of sexual violence is an instrument to guarantee rights and social protection that allows health, education, social assistance, Guardianship Councils, and justice professionals to adopt measures to care for the victims [24]. The Ministry of Health’s Ordinance 1271 of 2014 made sexual violence a grievance with mandatory immediate reporting (within 24 h), to ensure intervention in cases through the health sector [25], as an institutional obligation, and it is up to healthcare professionals to notify in accordance with current legislation [6], in order to introduce the adolescents into a protection and healthcare network.

In recent years in Brazil, studies that describe the occurrence and the characteristics of sexual violence and its victims have been published [2,7]. However, data regarding factors associated with the sexual violence, especially among those cases that are treated by health services and registered in SINAN remain scarce. Since the events that are attended to by health services may have specific characteristics, it is relevant to understand their particularities. In Espirito Santo state, there are still no records in the literature on the factors associated with this condition in adolescence. Hence, this study aimed to describe the prevalence of sexual violence against adolescents reported by health services from Espirito Santo, Brazil, which were registered in SINAN, and to analyze its associated factors.

## 2. Materials and Methods

### 2.1. Study Design and Population

An analytical cross-sectional study was carried out with all reported cases of sexual violence against adolescents (10 to 19 years old) in Espírito Santo state, Brazil, between 2011 and 2018. Espírito Santo is situated in southeastern Brazil, with 46,074.444 km^2^ of land area. According to the last census (IBGE), the state had about 3.5 million inhabitants, and the population of adolescents estimated at the time was 603,835, with a demographic density of 76.25 inhab./km^2^ and a Human Development Index (HDI) of 0.740 [26].

According to the World Health Organization (WHO) adolescence is the period of life that begins at age 10 and ends at age 19. For the WHO, adolescence is divided into two phases: pre-adolescence—from 10 to 14 years old and adolescence—from 15 to 19 years of age [27].

### 2.2. Procedures for Data Collection

In Brazil, interpersonal and self-inflicted violence is part of a continuous epidemiological surveillance system, which means every suspected or confirmed case of violence must be reported. To accomplish this requirement, the health professional responsible for assisting the victim must report the event to the authorities and other health services that will provide subsequent care [6].

The report is made by filling in the “Interpersonal/Self-inflicted Violence Notification Form”. This form is the same used in all the national territories of Brazil, settled by the Ministry of Health [6].

After the health professional fills in the form, a copy is sent to the Epidemiological Surveillance of the municipality, where the data are entered into SINAN. SINAN is one of the National Health Information Systems. It aims to collect, convey, and disseminate data generated by the Epidemiological Surveillance System to supporting the investigation process and provide subsidies for the analysis of information on compulsory reporting injuries, which includes transmissible disease and violence [28].

By filling in this form, two major purposes are reached: to signal to the healthcare network, comprising health, social, and law services, the needs of the victim and to feed the National Health Information System, SINAN [6].

As SINAN is a secondary database, an exploratory descriptive analysis was carried out following instructive reporting of interpersonal and self-inflicted violence guidelines in order to qualify the data. Duplicate cases were verified by organizing the records by the date of reporting, comparing the date of occurrence, the victim’s name, the date of birth, and the mother’s name, which were later excluded.

Initially, there were 1924 reported cases of sexual violence registered in SINAN. However, only 561 reporting forms were analyzed, which included those whose variables of interest for the study did not have the terms “ignored” or “blank”; therefore, the forms in which the investigated variables were missing were not included in the analysis.

### 2.3. Variables

The study’s dependent variable was sexual violence (no/yes).

The outcome of this study was being a suspect or a confirmed victim of sexual violence. According to the guidelines for filling in the individual notification form of violence, sexual violence is defined as: any action in which a person, taking advantage of his/her position of power and using physical force, coercion, intimidation, or psychological influence, with or without the use of weapons or drugs, compels another person, of any sex and age, to have, witness, or participate in any way in sexual interactions or to use their sexuality, for profit, revenge, or any other intention. It includes rape, incestuous abuse, sexual harassment, forced sex in marriage, nonconsensual, imposed sexual games and erotic practices, child pornography, pedophilia, voyeurism, handling, oral, anal, or genital penetration with a penis or objects in a forced way. It also includes coercive/embarrassing exposure to lewd acts, exhibitionism, masturbation, erotic language, sexual interactions of any kind, and pornographic material. Likewise, sexual violence is characterized by acts that, through coercion, blackmail, bribery, or solicitation, prevent the use of any contraceptive method or force marriage, pregnancy, abortion, or prostitution; or that limit or nullify the autonomy and exercise of the sexual and reproductive rights of any person.

The independent variables were victim characteristics, aggression, and perpetrator characteristics. Victim characteristics were categorized as follows: age (10 to 12 years, 13 to 17 years, 18 to 19 years); race/color (white/non-white—brown, yellow, indigenous); disability/disorder (no/yes); area of residence (urban/rural, peri-urban); place of occurrence: home (yes/no–collective housing, school, sports venue, bar or similar, public road, commerce/services, industries/construction, other); occurred other times (no/yes); number of perpetrators (one/two or more); perpetrator’s age (up to 19 years/20 years or older); perpetrator’s sex (male/female/both sexes); relationship with the victim (partners and ex-partners—spouse, ex-spouse, boyfriend, ex-boyfriend/parental relationship—father, mother, stepmother, stepfather/acquaintance, and others/unknown); suspected alcohol use (no/yes); and referral (no/yes).

### 2.4. Statistical Analysis

To obtain the prevalence ratios, adjustment was considered via the log-binomial regression model, which is the particular case of the generalized linear regression model when considering the response variable with the Bernoulli distribution and logarithmic link function. All statistics analysis were performed in the STATA version 15.1 (StataCorp, College Station, TX, USA), and the level of significance adopted was alpha fixed in 5%.

### 2.5. Ethical Considerations

The study was approved by the Institutional Review Board of the Federal University of Espírito Santo, under Opinion 2,819,597, and all norms and guidelines of Resolution 499/2012 of the National Health Council of Brazil were respected.

## 3. Results

From 2011 to 2018, 561 cases of sexual violence were reported, representing a frequency of 32.6% of cases of reporting violence against adolescents, of which 522 (93.0%) cases occurred among female adolescents and 39 (7.0%) among male adolescents.

Table 1 presents the victim characteristics. It is noted that, regardless of sex, most youth were between 13 and 17 years old, were non-white, did not have any disorders or disabilities, and lived in urban areas. Sexual violence mostly occurred at home, and victims reported a history of previous sexual violence. Regarding the perpetrator characteristics, in most cases, there was only one perpetrator, who was typically over 20 years old, male, an acquaintance of the victims, and without suspicion of alcohol use.

Table 2 describes the bivariate analyses. Sexual violence in both sexes was predicted by the act, place of occurrence, occurring other times, and the relationship with the victim (*p* < 0.05). In females, the prevalence of sexual violence was also related to disability/disorder, the number of perpetrators, the perpetrator’s sex, and a perpetrator with suspected alcohol use (*p* < 0.05).

The prevalence of cases of sexual violence against girls was 2.69 times (95%CI: 2.07–3.50) higher among those aged between 10 and 12 years compared to the group of girls aged 18 and 19 years. There was a higher occurrence of sexual violence at home than other places (PR: 1.57; 95%CI: 1.33–1.86), when there was a history of sexual violence (PR: 1.21; 95%CI: 1.08–1.36), and 1.58 times (95%CI: 1.28–1.95) by a lone perpetrator. As for the relationship with the victims, sexual violence against girls was approximately 1.9 times more perpetrated by strangers than by partners or ex-partners (e.g., boyfriends, girlfriends) (95% CI: 1.57–2.34) (Table 3).

In males, sexual violence was 3.47 times (95%CI: 1.01–11.92) higher among adolescents aged between 10 and 12 years compared to the occurrence among those aged between 18 and 19 years. It was associated with a higher prevalence of occurrences at home (PR: 4.44; 95%CI: 1.97–10.01) and with a history of sexual violence (PR: 3.56. 95%CI: 1.71–7.39). Perpetrators aged 20 years or more increased the prevalence of sexual violence against boys by approximately two times (95%CI: 1.03–4.02), and the outcome was practiced more by partners and ex-partners than by parents (PR: 0.12; 95%CI: 0.03–0.53) (Table 4).

## 4. Discussion

Among all the other types of violence registered by the health services in Espírito Santo, Brazil, it was observed that the prevalence of sexual violence against adolescents was 32.6%, higher than the previously reported national prevalence of 24.7% between 2011 and 2017 [29]. In 2021, in Espírito Santo, sexual violence was the most reported type of violence against adolescents of 10 to 14 years and the third among adolescents of 15 to 19 years, at 42.0% and 14.2%, respectively [30]. A study conducted in Madrid, Spain that described the clinical–epidemiological characteristics, management and followup of suspected child abuse diagnosed in Pediatric Emergencies, showed a prevalence of 40.6% of physical abuse, 35.1% of neglect, and 25.5% of sexual violence [31]. A similar prevalence of sexual violence was found in a cross-sectional study that identified 31.8% of reported physical violence, 36.8% of emotional violence, and 26.7% of sexual violence against adolescents [32].

Since SINAN gathers data from all Brazilian cities and states, the prevalence of the types of violence in the studies available in the literature overlaps. Nevertheless, these numbers are based on reported cases, and we need to consider that many events are not identified or reported by a health professional; therefore, their real occurrence may be underestimated. As for associated factors, sexual violence was significantly predicted by the younger age of the victim with those 10 to 12 years old most at risk. Most sexual violence cases occurred at home and were part of a history of violence with a perpetrator known to the victims. In females, the sexual violence was also more likely by a lone perpetrator, and in males, the perpetrators were likely to be aged 20 years and older.

The number of reported cases of sexual violence against girls was significantly higher when compared to the number of reported cases in male adolescents. This finding is similar to that in a study conducted with reported cases of sexual violence throughout Brazil, in which the prevalence of the outcome was 77.1% in females [30], similar to that found in other studies [33,34,35,36]. It is worth highlighting a study of cases of violence reported in Pernambuco where the prevalence of sexual violence was higher in females compared to males [37].

Historically, in the context of a patriarchal society deeply marked by the asymmetry of power in gender relations, there is a perpetuation of eroticization and objectification of the female body, which begins during childhood and intensifies during adolescence [38]. In this context, it is important to emphasize that sexual violence is one of the manifestations of gender inequality that mainly affects women across the lifespan and one of the cruelest forms of the demonstration of dominance imposed on them [2]. Sexual violence against adolescents is a favored form of gender-determined violence, as it is usually perpetrated by an older experienced man who has a trusting relationship with the victim [39,40].

Although sexual violence is mostly committed against girls, it is important to emphasize that it also affects male youth; however, the data available in the literature limit knowledge of its magnitude [1]. Some factors may be related to the underreporting of these cases in males, namely: fear of homosexuality and/or fear of being seen as homosexual; emotional responses differentiated from teenagers; fear of being blamed, as they are generally seen as able to defend themselves; the difficulty of guardians to perceive the relevant signs and symptoms of abuse; and the denial of abuse when it occurs through perpetrators such as parents and other adolescents [41,42].

In both sexes, there was a higher prevalence of reports of sexual violence against adolescents aged between 10 and 12 years. Girls become more exposed to sexual violence during puberty, when secondary sexual characteristics develop [1,34]. Moreover, individuals in the early stages of adolescence are physically, psychologically, and socially more vulnerable, not having sufficient maturity to understand or anticipate violence by the perpetrators, who often gain the victims’ trust and may also impose authority over victims to perpetrate the violence [2,34]. Therefore, in order to break this type of violence, it is necessary that third parties denounce and activate the protection network for adolescents, as they depend on the initiative of others to break the silence [2].

It is important to highlight the higher prevalence of sexual violence against girls in the age groups from 10 to 12 years and from 13 to 17 years, compared to the age of 18 to 19 years, suggesting perpetuation of the cycle of violence throughout adolescence. In males, the highest prevalence was associated only with boys aged between 10 and 12 years. These findings suggest that boys are victims more often while there is still no possibility of defense, and for this reason, the outcome is less prevalent in older ages [2].

In the present study, the main place of occurrence of sexual violence was in the victims’ homes. A survey of reported cases of sexual violence registered in SINAN observed that the chances of sexual violence were approximately twice as high at home than in other spaces [35]. In a study carried out in the state of Paraná, sexual violence was most often carried out in the homes of victims, followed by the homes of perpetrators and relatives [34].

The first experience of human interaction usually takes place at home, in a positive, nurturing, and loving context. However, the home is also a place where the first exposure to violence and perpetuation of the cycle is likely to occur [1]. Due to home privacy, violence against children and adolescents is practiced without public knowledge, making interventions difficult to carry out to deconstruct young people’s perception of a safe and trustworthy place [16,34,35,40]. In addition to these factors, home privacy contributes to the silence of recurrent episodes [35].

After adjusting for the multivariate analysis, revictimization remained associated with a higher prevalence of sexual violence in both sexes. In a longitudinal study carried out with adolescents in the USA, there was a recurrence of sexual violence in a quarter of victims during the academic semester [43]. Between 2010 and 2014, 680 cases of sexual violence against adolescents were reported in schools, of which 40.0% were recurrent cases [33], supporting similar prevalence found in other studies conducted with data from SINAN [35,44].

In a study that analyzed the recurrence of reported cases of violence, it was possible to note that there was a recurrence of sexual violence or negligence, even after previous reports of physical and sexual violence, suggesting that in some cases reporting is not enough to change the victim’s environment may not prevent revictimization [45]. A systematic review revealed that revictimization was associated with greater suffering, the presence of psychiatric disorders, difficulty with interpersonal relationships and coping, self-blame, shame, and revictimization in adult life [46,47,48,49].

Regarding perpetrator characteristics, a higher prevalence of sexual violence among females remained associated with having only one perpetrator. According to national descriptive studies that used reported data on violence, most sexual assaults against adolescents are by a lone perpetrator [29,35]; however, it is important to emphasize that although the reported cases mostly contain only one perpetrator, adolescents may be exposed to more than one perpetrator frequently [43].

In males, the perpetrator was likely to be an adult (aged 20 years or more) regardless of R adjustment. This finding corroborates other research in which most perpetrators were adults [50]. The relationship of adults with children and adolescents is guided by hierarchy and power relations; thus, adults can more easily take advantage of these relationships to coerce, manipulate, and attack their victims who are more vulnerable [51].

As for the perpetrators’ relationships with the victims, the highest prevalence of sexual violence was associated with the perpetrator being unknown. During adolescence, contact with people in environments outside family life tends to increase, which makes adolescents more exposed to violence by people outside the home. However, it is worth raising the hypothesis that suspected or confirmed cases of sexual violence committed by strangers or by acquaintances without family ties to the victim can be more easily reported by youth and relatives, compared to cases where the perpetrator is a relative, since chronicity and silence are common in the domestic setting [39]. Additionally, it is noteworthy that the data in this study came from the reporting of the health sector, which can facilitate the omission by victims and their families of the perpetrator’s true identity [2].

### 4.1. Implications

Such findings can contribute to the development of public health policies to advance the fight against this problem in an interdisciplinary scope. Hence, healthcare, social assistance, public security, and education must work together with prevention, management, and biopsychosocial recovery actions in order to promote comprehensive assistance to victims as well as broaden our view of adolescents, a group vulnerable to violence.

### 4.2. Limitations

The limitations of this study included the use of a secondary database, which makes it impossible to correct, or fill in, missing data. Furthermore, selection bias may also be a limitation as only cases reported via the health system were reported; that is, we did not access legal system reports. Thus, it is not possible to infer the real prevalence of sexual violence, since there are cases that, even when they arrive at the health service, are not reported, and cases that occur in the community in general that are not attended to by health services.

Despite the limitations, studies using secondary databases such as SINAN are important for determining the predictors and profiles of sexual violence for healthcare professionals and managers as well as stakeholders. This type of study is also relevant to signal managers about possible improvements that can be made in the sexual violence reporting instrument, in information systems, and in the continuing education of professionals about the identification and reporting of injuries.

## 5. Conclusions

In order to prevent sexual violence, it is relevant to have available studies of its prevalence and characteristics conducted in the general community and in specific scenarios, such as in health services. Hence, identifying the prevalence of reported cases of sexual violence by health services in Espirito Santo, Brazil, and its associated factors are the main contributions of this research.

Given the high magnitude and complexity of this injury throughout the Brazilian territory, carrying out studies that assess the distribution and factors associated with sexual violence at the municipal and state levels are relevant in order to assess the systemic weaknesses and intervene adequately, responding to situations according to local realities.

It is important to consider the need to approach and work on sexual education with adolescents so that they can know their rights, recognize possible situations of sexual violence, and expose such experiences in order to be protected. Schools are fundamental to fostering such age-appropriate discussions and to promoting an environment where: (1) students can expose situations in which their rights have been violated, and (2) students feel safe and protected.

It is hoped that the reporting of cases of sexual violence against youth can be encouraged, contributing to the development and improvement of public policies to promote health and provide safety for youth. National studies with primary data should be conducted to strengthen enable greater understanding and the creation of policies, programs, and intervention strategies that can be carried out in local contexts.

## Figures and Tables

**Table 1 ijerph-19-14481-t001:** Characteristics of reported cases of sexual violence against adolescents aged 10 to 19 years. Espírito Santo, Brazil, 2011 to 2018 (*n* = 561).

Variables	Female (*n* = 522)			Male (*n* = 39)		
	*n*	(%)	95%CI	*n*	(%)	95% CI
Age group						
10 to 12 years	169	32.4	28.50–36.51	17	43.6	29.30–59.02
13 to 17 years	300	57.5	53.19–61.64	19	48.7	33.87–63.80
18 to 19 years	53	10.2	7.85–13.04	03	7.7	2.65–20.32
Race/color						
White	137	26.1	22.47–29.98	06	15.4	7.25–29.73
Non-white	386	73.9	70.02–77.53	33	84.6	70.27–92.75
Disability/Disorder						
No	481	92.1	89.52–94.16	33	84.6	70.27–92.75
Yes	41	7.9	5.84–10.48	06	15.4	7.25–29.73
Area of residence						
Urban	451	86.4	83.19–89.07	35	89.7	76.42–95.94
Rural/Peri-urban	71	13.6	10.93–16.81	04	10.3	4.06–23.58
Place of occurrence: Home						
No	113	21.6	18.33–25.38	08	20.5	10.78–35.53
Yes	409	78.4	74.64–81.67	31	79.5	64.47–89.22
Occurred other times						
No	219	42.0	37.79–46.23	10	25.6	14.57–41.08
Yes	303	58.0	53.77–62.21	29	74.4	58.92–85.43
Number of perpetrators						
One	459	87.9	84.86–90.45	32	82.1	67.33–91.02
Two or more	63	12.1	9.55–15.14	07	17.9	8.98–32.67
Perpetrator’s age (years)						
Up to 19 years	122	23.4	19.94–27.19	08	20.5	10.78–35.53
20 years and older	400	76.6	72.81–80.06	31	79.5	64.47–89.22
Perpetrator’s sex						
Both sexes	07	1.3	0.65–2.74	01	2.6	0.45–13.18
Female	07	1.3	0.63–2.74	04	10.3	4.06–23.58
Male	508	97.4	95.57–98.40	34	87.1	73.29–94.40
Relationship with the victim						
Partners and ex-partners	77	14.8	11.97–18.05	02	5.1	1.42–16.89
Parental relationship	147	28.2	24.47–32.17	04	10.3	4.06–23.58
Acquaintance/others	219	41.9	37.79–46.23	27	69.2	53.58–81.43
Unknown	79	15.1	12.32–18.46	06	15.4	7.25–29.73
Suspected alcohol use						
No	395	75.7	71.81–79.16	30	76.9	61.66–87.35
Yes	127	24.3	20.84–28.19	09	23.1	12.65–38.34
Referral						
No	49	9.4	7.17–12.19	02	5.1	1.42–16.89
Yes	473	90.6	87.81–92.83	37	94.9	83.11–98.58

**Table 2 ijerph-19-14481-t002:** Distribution of the characteristics of the reports of sexual violence among adolescents by sex Espírito Santo, Brazil, 2011 to 2018 (*n* = 561).

Variables	Female (*n* = 522)				Male (*n* = 39)			
	*n*	(%)	95% CI	*p*-Value	*n*	(%)	95% CI	*p*-Value
Age group								
10 to 12 years	169	65.5	59.51–71.04	0.000	17	14.9	9.52–22.59	0.013
13 to 17 years	300	41.7	38.12–45.30		19	7.3	4.73–11.13	
18 to 19 years	53	18.2	14.20–23.05		03	3.8	1.28–10.45	
Race/color								
White	137	40.8	35.70–46.20	0.899	06	4.9	2.27–10.32	0.091
Non-white	386	41.2	38.13–44.42		33	9.9	7.17–13.63	
Disability/Disorder								
No	481	40.3	37.54–43.09	0.014	33	8.2	5.92–11.33	0.435 (*)
Yes	41	54.7	43.45–65.43		06	11.3	5.29–22.58	
Area of residence								
Urban	451	40.5	37.64–43.40	0.207	35	9.2	6.72–12.57	0.271
Rural/Peri-urban	71	45.8	38.16–53.66		04	5.3	2.09–12.93	
Place of occurrence: Home								
No	113	28.2	24.00–32.77	0.000	08	3.5	1.77–6.68	0.000
Yes	409	47.1	43.82–50.45		31	13.9	9.97–19.06	
Occurred other times								
No	219	36.0	32.25–39.85	0.000	10	3.7	2.02–6.68	0.000
Yes	303	45.9	42.14–49.72		29	15.8	11.20–21.72	
Number of perpetrators								
One	459	44.3	41.27–47.30	0.000	32	10.3	7.38–14.16	0.057
Two or more	63	27.2	21.84–33.22		07	4.9	2.39–9.76	
Perpetrator’s age (years)								
Up to 19 years	122	39.7	34.42–45.31	0.568	08	6.1	3.10–11.50	0.219
20 and older	400	41.6	38.51–44.72		31	9.6	6.87–13.34	
Perpetrator’s sex								
Both sexes	07	10.6	5.23–20.31	0.000	01	2.9	0.52–14.92	0.262 (*)
Female	07	2.9	1.40–5.80		04	5.2	2.04–12.61	
Male	508	53.0	49.81–56.11		34	9.9	7.18–13.53	
Relationship with the victim								
Partners and ex-partners	77	24.0	19.64–28.95	0.000	02	11.8	3.29–34.34	0.014 (*)
Parental relationship	147	44.1	38.91–49.51		04	2.9	1.12–7.17	
Acquaintance/others	219	45.9	41.49–50.40		27	11.3	7.91–16.00	
Unknown	79	57.3	48.91–65.19		06	10.0	4.66–20.15	
Suspected alcohol use								
No	395	43.8	40.63–47.10	0.002	30	9.0	6.34–12.50	0.642
Yes	127	34.5	29.84–39.51		09	7.6	4.03–13.75	
Referral								
No	49	36.8	29.12–45.30	0.288	02	3.8	1.04–12.75	0.293 (*)
Yes	473	41.6	38.80–44.53		37	9.2	6.77–12.46	

(*) Fisher’s exact test.

**Table 3 ijerph-19-14481-t003:** Raw and adjusted analysis of the effects of the characteristics of cases of sexual violence against female adolescents. Espírito Santo, Brazil, 2011–2018 (*n* = 522).

Variables	Raw Analysis			Adjusted Analysis (**)		
	PR	95% CI	*p*-Value	PR	95% CI	*p*-Value
Age group						
10 to 12 years	3.60	2.78–4.66	0.000	2.69	2.07–3.50	0.000
13 to 17 years	2.29	1.77–2.96		2.05	1.58–2.64	
18 to 19 years	1.0	--		1.0	--	
Race/color						
White	1.0	--	0.899	1.0	--	0.985
Non-white	1.01	0.87–1.17		0.99	0.93–1.15	
Disability/Disorder						
No	1.0	--	0.014	1.0	--	0.176
Yes	1.36	1.09–1.69		1.17	0.93–1.46	
Area of residence						
Urban	0.88	0.73–1.06	0.207	0.92	0.77–1.09	0.312
Rural/Peri-urban	1.0	--		1.0	--	
Place of occurrence: Home						
No	1.0	--	0.000	1.0	--	0.000
Yes	1.67	1.41–1.98		1.57	1.33–1.86	
Occurred other times						
No	1.0	--	0.000	1.0	--	0.001
Yes	1.28	1.12–1.46		1.21	1.08–1.36	
Number of perpetrators						
One	1.63	1.31–2.03	0.000	1.58	1.28–1.95	0.000
Two or more	1.0	--		1.0	--	
Perpetrator’s age (years)						
Up to 19 years	1.0	--	0.568	1.0	--	0.720
20 and older	1.05	0.89–1.22		1.03	0.89–1.18	
Relationship with the victim						
Partners and ex-partners	1.0	--	0.000	1.0	--	0.000
Parental relationship	1.84	1.46–2.31		1.25	1.01–1.54	
Acquaintance/others	1.91	1.54–2.38		1.70	1.39–2.07	
Unknown	2.39	1.87–3.04		1.92	1.57–2.34	
Suspected alcohol use						
No	1.0	--	0.002	1.0	--	0.271
Yes	0.79	0.67–0.92		0.92	0.79–1.06	

(**) Log-binomial model.

**Table 4 ijerph-19-14481-t004:** Crude and adjusted analysis of the effects of the characteristics of cases of sexual violence against male adolescents. Espírito Santo, Brazil, 2011–2018.

Variables	Raw Analysis			Adjusted Analysis (**)		
	PR	95% CI	*p*-Value	PR	95% CI	*p*-Value
Age group						
10 to 12 years	3.98	1.21–13.12	0.013	3.47	1.01–11.92	0.037
13 to 17 years	1.95	0.59–6.42		2.34	0.69–7.89	
18 to 19 years	1.0	--		1.0	--	
Race/color						
White	1.0	--	0.091	1.0	--	0.982
Non-white	2.02	0.87–4.70		1.01	0.51–1.97	
Disability/Disorder						
No	1.0	--	0.435 (*)	1.0	--	0.377
Yes	1.38	0.61–3.13		1.23	0.78–1.95	
Area of residence						
Urban	1.73	0.63–4.73	0.271	1.03	0.42–2.56	0.942
Rural/Peri-urban	1.0	--		1.0	--	
Place of occurrence: Home						
No	1.0	--	0.000	1.0	--	0.000
Yes	4.01	1.89–8.54		4.44	1.97–10.01	
Occurred other times						
No	1.0	--	0.000	1.0	--	0.000
Yes	4.26	2.13–8.52		3.56	1.71–7.39	
Number of perpetrators						
One	2.10	0.95–4.65	0.057	1.61	0.77–3.35	0.239
Two or more	1	--		1	--	
Perpetrator’s age (years)						
Up to 19 years old	1.0	--	0.218	1.0	--	0.020
20 and older	1.59	0.75–3.36		2.04	1.03–4.02	
Relationship with the victim						
Partners and ex-partners	1.0	--	0.018	1.0	--	0.000
Parental relationship	0.24	0.05–1.24		0.12	0.03–0.53	
Acquaintance/others	0.96	0.25–3.72		1.61	0.49–5.29	
Unknown	0.85	0.19–3.84		2.17	0.56–8.42	
Suspected alcohol use						
No	1.0	--	0.642	1.0	--	0.937
Yes	0.84	0.41–1.73		1.01	0.55–1.82	

(*) Fisher’s exact test; (**) Log-binomial model.

## Data Availability

Not applicable.

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
