# Peer review of "Sexual Violence against Adolescents in the State of Espírito Santo, Brazil: An Analysis of Reported Cases"

_ijerph, 2022, doi:10.3390/ijerph192114481_

Round 1

Reviewer 1 Report

Summary

Thank you for the opportunity to review, “Sexual violence against adolescents in Brazil: an analysis of reported cases.” This study presents prevalence estimates and associated factors related to adolescent sexual abuse. They reported that the prevalence of sexual abuse was 33%, and among victims, 93% were female. In general, being younger, living at home, being related to the perpetrator and experiencing prior sexual abuse was all associated with sexual abuse victimization.

This manuscript is concise, clear, and well-written. The subject matter is of relevance, both to the field and the IJERPH readership. Further, the authors did a good job contextualizing their results into the broader literature.

If this paper is to advance, I suggest the authors consider the following questions/comments:

1.     Contribution/Originality/Novelty: The introduction is generally well written and provides a nice overview of adolescent sexual abuse statistics. I think what is missing is a discussion of gaps in the existing research and how this particular study is situated to address those gaps. The authors should directly state how this study advances knowledge on sexual abuse (in general) or sexual violence in Brazil specifically. Then, the authors should return to this contribution in the discussion/conclusion because after reading, I still was not exactly sure how this study advances what we know about adolescent sexual violence in Brazil.

2.     Missing data: The authors state that during the early stages of analysis, they removed cases in which key variables were missing or ignored. What proportion of the total cases were removed for this reason? This information might be helpful to present.

3.     Clarification on Prevalence: Regarding the 33% of cases of violence involving sexual violence, do the authors have any sense of what that means for the prevalence of sexual violence against adolescents generally? Is there any work that looks at the coverage of SINAN, or estimates of sexual violence in the population?

4.     Interpretation of prevalence: Related to the question above, I also had a question about the interpretation of the prevalence estimates. On page 14 (line 316), the authors write, “This study revealed the high prevalence sexual violence among youth…” Is that exactly accurate? Wasn’t this the prevalence of sexual violence among reports of youth violence? Perhaps this should be clarified.

5.     Dependent variable: How is sexual violence defined in SINAN? What types of offenses does it include? Is it just contact offenses, or is harassment included too? It would also be helpful to know how sexual violence was defined in the other prevalence studies cited by the authors.

6.     English language/style: One of the journal’s review items asks about English language/style. There are several instances of this use of hyphens that are not needed. These can be corrected with a minor proofread.

Author Response

Reviewer 1

Comments and Suggestions for Authors

Thank you for the opportunity to review, “Sexual violence against adolescents in Brazil: an analysis of reported cases.” This study presents prevalence estimates and associated factors related to adolescent sexual abuse. They reported that the prevalence of sexual abuse was 33%, and among victims, 93% were female. In general, being younger, living at home, being related to the perpetrator and experiencing prior sexual abuse was all associated with sexual abuse victimization.

This manuscript is concise, clear, and well-written. The subject matter is of relevance, both to the field and the IJERPH readership. Further, the authors did a good job contextualizing their results into the broader literature.

If this paper is to advance, I suggest the authors consider the following questions/comments:

1. Contribution/Originality/Novelty: The introduction is generally well written and provides a nice overview of adolescent sexual abuse statistics. I think what is missing is a discussion of gaps in the existing research and how this particular study is situated to address those gaps. The authors should directly state how this study advances knowledge on sexual abuse (in general) or sexual violence in Brazil specifically. Then, the authors should return to this contribution in the discussion/conclusion because after reading, I still was not exactly sure how this study advances what we know about adolescent sexual violence in Brazil.
Response: Thank you very much for your careful review. We have added these information as well as clarification on the introduction and conclusion section as per recommended.
2. Missing data: The authors state that during the early stages of analysis, they removed cases in which key variables were missing or ignored. What proportion of the total cases were removed for this reason? This information might be helpful to present.
Response: Thank you for this comment and suggestion. Added in the method. Page 4, lines 179-190.

3. Clarification on Prevalence: Regarding the 33% of cases of violence involving sexual violence, do the authors have any sense of what that means for the prevalence of sexual violence against adolescents generally? Is there any work that looks at the coverage of SINAN, or estimates of sexual violence in the population?
Response: The authors added this information in the discussion. Page 13. Lines 346-363.

4. Interpretation of prevalence: Related to the question above, I also had a question about the interpretation of the prevalence estimates. On page 14 (line 316), the authors write, “This study revealed the high prevalence sexual violence among youth…” Is that exactly accurate? Wasn’t this the prevalence of sexual violence among reports of youth violence? Perhaps this should be clarified.
Response: We rewrote this information to make clearer and consistent throughout the text

5.     Dependent variable: How is sexual violence defined in SINAN? What types of offenses does it include? Is it just contact offenses, or is harassment included too? It would also be helpful to know how sexual violence was defined in the other prevalence studies cited by the authors.
Response: Thank you for your valuable commentary. We have  added this information in the method section. Page 4, lines 206-223.

6.     English language/style: One of the journal’s review items asks about English language/style. There are several instances of this use of hyphens that are not needed. These can be corrected with a minor proofread.
Response: It was an error in Microsoft Word when we pasted the text. It was corrected.

Reviewer 2 Report

The manuscript presents an important issue, however it does not bring any scientific novelty for the understanding of the sexual violence among adolescents in Brazil. The method is also quite simple and the conclusions are already recognized in the literature. For those reasons aforementioned I reject the paper for publication in the IJERPH.

Author Response

Response: This is a very limited and misguided analysis by this reviewer. Furthermore, it is unsubstantiated and unsupported.
We have a clear research question, with a clear objective, reproducible methodology and appropriate analyses as pointed out by the other two reviewers and the Associate Editor. This opinion is strange. Could you be clearer with these statements of yours?

Reviewer 3 Report

Sexual violence against adolescents in Brazil: an analysis of reported cases

This is an interesting and important contribution, with relevant implications to the field. Nevertheless, I believe that a few changes would improve the overall quality of the paper:

1.     Title: is the state of Espírito Santo representative of Brazil?

2.     Abstract: please include info regarding the measurement instruments used; also, replace population with sample (line 24).

3.     Line 36: “Adolescence is a stage of extreme vulnerability to sexual violence”. Why?

4.     Please include a definition of sexual violence, including its typology;

5.     Please include more results of similar studies in other countries, focusing on the particularities of Brazil in general, and ES in particular.

6.     Line 93: authors must describe objectives with greater detail.

7.     How was the sample size calculated?

8.     Inclusion/exclusion criteria?

9.     Ethical considerations must include informed consent, confidentiality, etc. Also, how was the authorization form parents obtained?

10.  How was sexual violence measured? What was asked to adolescents?

11.  Please include an implications sections, focusing on social, political and health policies to prevent sexual violence against adolescents in Brazil, addressing cultural particularities of this culture.

Best wishes.

Author Response

Reviewer 3:

1.     Title: is the state of Espírito Santo representative of Brazil?
Response: The authors agree with the placement and, considering Espírito Santo, is a state of Brazil, which is part of the southeastern region of the country, and that Brazil is a continental country, with a territorial and heterogeneous dimension, being important to carry out studies in their different states, especially when it comes to a complex and multicausal theme such as violence. Knowing the data of each location is essential, so the title was modified, including the name of Espírito Santo state, Brazil where it was performed.
2.     Abstract: please include info regarding the measurement instruments used; also, replace population with sample (line 24).
Response: In the summary and methodology, the authors provide information about the system where data on sexual violence against adolescents was collected, the collection took place in the Information System on Notifiable Diseases, SINAN. In this sense, the authors did not use a collection instrument and the studied population consisted of all registered cases.
3.   Line 36: “Adolescence is a stage of extreme vulnerability to sexual violence”. Why?
Response: We have excluded the word “extreme”. The reasons for why adolescence is a stage of vulnerability to sexual violence, is written throughout the introduction. 
4. Please include a definition of sexual violence, including its typology;
Response: We’ve added this information in the method section. Line 148.
5. Please include more results of similar studies in other countries, focusing on the particularities of Brazil in general, and ES in particular.
Response: Added this information in the discussion section. Line 239.
6.     Line 93: authors must describe objectives with greater detail.
Response: We have added more information in the objective. 
7.  How was the sample size calculated?
Response: Thank you for this comment. In fact, as described in the method, it was worked with all cases of sexual violence against adolescents registered in Espírito Santo in from 2011 to 2018, not being worked with a sample but the population, thus, there is no calculation sample.
8.     Inclusion/exclusion criteria?
Response: We appreciated the careful revision. This information is flagged in the method. (“Duplicate cases were verified by organizing the records by date of reporting, comparing date of occurrence, victim’s name, date of birth and mother’s name, and were later excluded. Furthermore, only reporting forms whose variables of interest in the study do not have the terms “ignored” or “blank” were analyzed.”)
9.     Ethical considerations must include informed consent, confidentiality, etc. Also, how was the authorization form parents obtained?
Response: Ethical approved was obtained by the ethics and research committee, as described in the method. As for the consent form, as stated in the method, the data were obtained from an information system and not directly from the adolescent, or, parents.
10.  How was sexual violence measured? What was asked to adolescents?
Response: Information added to the method, dichotomized (yes/no), and as described in the methodology, the data were obtained from a violence notification form, where the question is whether the victim was a victim of violence, yes or no. This form is filled out by health professionals, with data on cases of victimized adolescents, whether suspected or confirmed. 
11.  Please include an implications sections, focusing on social, political and health policies to prevent sexual violence against adolescents in Brazil, addressing cultural particularities of this culture.
Response: We have added this information. Line 355. 

All co-authors have agreed to the resubmission with these revisions. The manuscript is consistent with the Guidelines for Authors of the IJERPH.
We are looking forward to hearing from you soon, and we hope to have our manuscript accepted for publication in this renowned journal.
Yours Sincerely,
The authors

Round 2

Reviewer 1 Report

IJERPH-1900946--Revision

(New Comments marked by *** under each author response)

1. Contribution/Originality/Novelty: The introduction is generally well written and provides a nice overview of adolescent sexual abuse statistics. I think what is missing is a discussion of gaps in the existing research and how this particular study is situated to address those gaps. The authors should directly state how this study advances knowledge on sexual abuse (in general) or sexual violence in Brazil specifically. Then, the authors should return to this contribution in the discussion/conclusion because after reading, I still was not exactly sure how this study advances what we know about adolescent sexual violence in Brazil.

Response: Thank you very much for your careful review. We have added these information as well as clarification on the introduction and conclusion section as per recommended.

***The authors sufficiently addressed this comment.

2. Missing data: The authors state that during the early stages of analysis, they removed cases in which key variables were missing or ignored. What proportion of the total cases were removed for this reason? This information might be helpful to present.

Response: Thank you for this comment and suggestion. Added in the method. Page 4, lines 179-190.

***In the revision, the authors state that only 561 reporting forms whose variables of interest in the study do not have the terms “ignored” or “blank” were analyzed. Does that mean that if any of the variables was missing, then the form was not included? Or, some X% of variables missing?

3. Clarification on Prevalence: Regarding the 33% of cases of violence involving sexual violence, do the authors have any sense of what that means for the prevalence of sexual violence against adolescents generally? Is there any work that looks at the coverage of SINAN, or estimates of sexual violence in the population?

Response: The authors added this information in the discussion. Page 13. Lines 346-363.

***The authors sufficiently addressed this comment.

4. Interpretation of prevalence: Related to the question above, I also had a question about the interpretation of the prevalence estimates. On page 14 (line 316), the authors write, “This study revealed the high prevalence sexual violence among youth…” Is that exactly accurate? Wasn’t this the prevalence of sexual violence among reports of youth violence? Perhaps this should be clarified.

Response: We rewrote this information to make clearer and consistent throughout the text

***The authors sufficiently addressed this comment.

5.     Dependent variable: How is sexual violence defined in SINAN? What types of offenses does it include? Is it just contact offenses, or is harassment included too? It would also be helpful to know how sexual violence was defined in the other prevalence studies cited by the authors.

Response: Thank you for your valuable commentary. We have  added this information in the method section. Page 4, lines 206-223.

***The authors sufficiently addressed this comment.

6.     English language/style: One of the journal’s review items asks about English language/style. There are several instances of this use of hyphens that are not needed. These can be corrected with a minor proofread.

Response: It was an error in Microsoft Word when we pasted the text. It was corrected.

***The authors sufficiently addressed this comment.

Author Response

Dear Reviewer,

***The authors sufficiently addressed this comment.

  1. Missing data: The authors state that during the early stages of analysis, they removed cases in which key variables were missing or ignored. What proportion of the total cases were removed for this reason? This information might be helpful to present.

    Response: Thank you for this comment and suggestion. Added in the method. Page 4, lines 179-190.

    ***In the revision, the authors state that only 561 reporting forms whose variables of interest in the study do not have the terms “ignored” or “blank” were analyzed. Does that mean that if any of the variables was missing, then the form was not included? Or, some X% of variables missing?

RESPONSE: Thank you for your review. Yes, it means that if any of the variables was missing, then the form was not included. We addressed this comment in page 4, line 144-46.

Reviewer 3 Report

Thank you for implementing all the requested changes to the manuscript. I believe they have very improved the overall quality of the article, and it is now fit for publication.

Best wishes.

Author Response

Response: Many Thanks!